# Cardio-pulmonary Substructure Segmentation of CT images using Convolutional Neural Networks for Clinical Outcome Analysis

**Rabia Haq**[1]                                                                HAQR@MSKCC.ORG
**Alexandra Hotca**[1]
**Aditya Apte**[1]
**Andreas Rimner**[2]
**Joseph O. Deasy**[1]
**Maria Thor**[1]

[1] *Department of Medical Physics, Memorial Sloan Kettering Cancer Center, New York, US*

[2] *Department of Radiation Oncology, Memorial Sloan Kettering Cancer Center, New York, US*

**Editors:** Under Review for MIDL 2019

**Keywords:** Semantic segmentation, Convolutional Neural Networks, cardio-pulmonary, clinical outcomes

## 1. Introduction

Various studies have shown that doses to some cardio-vascular substructures may be critical factors in the observed heart toxicity and early mortality following radiotherapy (RT) for nonsmall cell lung cancer (NSCLC) patients (Vivekanandan et al., 2017; McWilliam et al., 2017; RT et al., 2017; Thor et al., 2018). This may be attributed to irradiation of particular constituents of the cardio-pulmonary system [2-5]. Currently, segmentation of cardio-pulmonary organs other than the whole heart and lung has been overlooked, and only these two organs are routinely defined as part of the treatment planning process. RT planning requires robust and accurate segmentation of organs-at-risk in order to maximize radiation to the disease location and spare the normal tissue as much as possible. The introduction of a new set of organs puts requirements on both segmentation accuracy and segmentation time that would result in an overhead of several hours of manual segmentation and contour refinement in the clinic.

To facilitate this, we built and validated a multi-label Deep Learning Segmentation (DLS) framework for accurate auto-segmentation of cardio-pulmonary substructures. The DLS framework utilized a deep convolutional neural network architecture to segment 12 cardio-pulmonary substructures (Feng et al., 2010) from Computed Tomography (CT) scans of 217 patients previously treated with thoracic RT. The segmented substructures are: Heart, Pericardium, Atria, Ventricles, Descending Aorta (DA), Left Atrium (LA), Right Atrium (RA), Left Ventricle (LA), Right Ventricle (RV), Inferior Vena Cava (IVC), Superior Vena Cava (SVC) and Pulmonary Artery (PA). We evaluate our framework using a hold-out dataset of 24 CT scans by calculating volumetric-based as well as dose-volume histogram (DVH) based validation metrics. The proposed model reduces substructure seg-

mentation time for a new patient from about one hour of manual segmentation to approximately 10 seconds. We demonstrate that the model is robust against variability in image quality characteristics, including the presence/absence of contrast. Resulting segmentation accuracy was judged adequate for extracting dose-volume histogram information for patient outcomes analyses following RT, with no statistical difference discovered between auto-generated and expert contour evaluation metrics.

## 2. Methods

Experimental data consisted of computerized tomography (CT) scans of 241 patients obtained from our institutional clinic. This data consisted of contrast as well as non-contrast enhanced images of varying imaging quality and resolution across different scanners. Manual expert segmentation for 12 organs-at-risk cardio-pulmonary structures was considered ground truth and used for model training, testing and validation.

Our approach leverages the deep neural network architecture of (Chen et al., 2018). Convolutional neural networks (CNNs) and encoder-decoder neural networks have been successfully employed for medical image segmentation tasks (Isensee et al., 2018), (Oktay et al., 2018), (Jin et al., 2018), (Oktay et al., 2018). The Deeplab encoder-decoder network architecture with atrous separable convolutions consists of spatial pyramid pooling that encodes multi-scale contextual information to capture spatial anatomical information of contiguous structures. Dense feature maps extracted in the last encoder network path consist of detailed semantic information. The decoder network is able to robustly recover structure boundaries through bilinear upsampling at a factor of 4 while applying atrous convolutions to reduce features before semantic labeling. We trained the network using ResNet-101(He et al., 2016) as the encoder network backbone with learning rate = 0.01 using policy learning rate scheduler (Liu et al., 2015), crop size=$513 \times 513$, batch size = 8, loss = cross-entropy, output stride = 16 for 50 epochs for dense label prediction. Our approach has been implemented using the Pytorch DL framework.

We quantitatively evaluated the auto-generated segmentations by comparing the DSC Score and 95th Percentile Hausdorff Distance (HD95 (mm)) of 24 patients against expert clinical segmentations. Additionally, we calculated the difference in RT DVH metrics between auto-generated and expert contours. The wilcoxon rank-sum test was performed to determine any statistical difference between the metrics.

## 3. Experiments and Results

Figure 1 displays the DSC Score results for 24 hold-out validation CT images. Our achieved DSC accuracies are comparable to the state-of-the-art multi-atlas (Luo et al., 2019) and deep learning methods (Dormer and et al., 2018) for segmenting cardio-pulmonary substructures from CT images.The highest segmentation accuracy was observed for the heart (median DSC = 0.96, median HD95= 3.48 $mm$), while the remaining structures achieving median accuracy ($0.81 \leq$ DSC $\leq 0.94$) and ($6\ mm \leq$ HD95 $\leq 3\ mm$), with highest HD95 surface distance accuracy observed for DA.

Table 1 displays the percentage difference between the DVH metrics and their associated P-values for six substructure contours that are found to be critical for determining heart

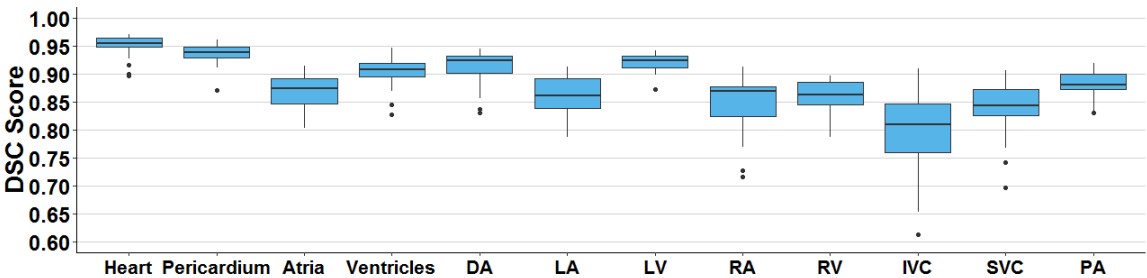

Figure 1: Median Dice Similarity Coefficient (DSC) Score results of 24 thoracic RT CT images comparing auto-generated DLS contours against manually segmented expert contours for 12 cardio-pulmonary sub-structures.

toxicity. These metrics were calculated using the RT treatment plan generated by physicists for each patient. None of the structure DVH metric differences were found to be statistically significant, with all P-values $> 0.05$. This indicates that the auto-generated segmentations are clinically acceptable for outcome analysis and RT treatment planning in the clinic.

| Structure (metric) | DLS to Expert Contour Difference (%) | P-value |
|---|---|---|
| Atria D45% (Gy) | 0.00 (-5.19  8.86) | 0.89 |
| Heart V2 (%) | 0.00 (0.00  0.01) | 0.93 |
| Heart V50 (%) | 0.00 (-0.05  0.10) | 0.83 |
| Left Atrium Dmax (Gy) | 0.00 (-0.28  0.31) | 0.90 |
| Left Atrium V63-V69 (%) | -0.06 (-0.25  0.75) | 0.92 |
| Pericardium MOH55% (Gy) | 1.76 (0.24  4.10) | 0.69 |
| Superior Vena Cava D90% (Gy) | 0.00 (-1.51  0.09) | 0.83 |
| Ventricles MOH5% (Gy) | -0.14 (-4.53  1.66) | 0.98 |

Table 1: Comparing Dose Volume Histogram (DVH) metrics of auto-generated DLS contours against Expert contours for 6 structures. Median and inter-quartile range of percentage differences between the two contours is presented, which is calculated as: $(DLS\_Volume - Expert\_Volume/Expert\_Volume) \times 100$.

## 4. Conclusion

We propose a model for auto-segmentation of cardio-pulmonary substructures from contrast and non-contrast enhanced CT images. We validated our approach by quantitatively comparing resulting contours against expert delineation, and further demonstrated no statistical difference when used for dose-volume histogram calculations. Resulting segmentations can effectively be utilized to study the effect of heart toxicity and clinical outcomes, as well as used as input to RT treatment planning. We have applied our approach to auto-segment additional 283 treatment planning CT scans to study heart toxicity outcomes in non-advanced lung patients. The developed cardio-pulmonary segmentation models are being integrated into deep learning tools within the open-source CERR (Deasy et al., 2003) platform.

## Acknowledgments

This research is partially supported by NCI R01 CA198121.

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
