# OpenReview forum: "Cardio-pulmonary Substructure Segmentation of CT images using Convolutional Neural Networks for Clinical Outcome Analysis"
_MIDL.io/2019/Conference/Abstract — MIDL Abstract 2019_

### Official Review · AnonReviewer1 · 2019-04-29
**Well-written abstract with good validation and results, limited methodological novelty**

**Rating:** 4
**Confidence:** 3

**Review:**

The abstract present results on segmentation of several cardiac and pulmonary structures using a DeepLab based algorithm. The results are competitive with the state of the arts and compared against expert segmentation. The abstract is well-written and the results presented adequately. Main criticism is the limited novelty in the methodology.

---

### Official Review · AnonReviewer2 · 2019-05-01
**accept if there is space**

**Rating:** 3
**Confidence:** 3

**Review:**

The paper abstract is discussing an relevant and important problem. The proposed approach seems sensible. The initial experimental results also look reasonable.

---

### Decision · Program_Chairs · 2019-05-06
**Acceptance Decision**

Accept